# Modeling the Lassa fever outbreak synchronously occurring with cholera and COVID-19 outbreaks in Nigeria 2021: A threat to Global Health Security

**Nancy B. Tahmo**[1☯]*, **Frankline Sevidzem Wirsiy**[1,2,3☯], **David M. Brett-Major**[1]

**1** University of Nebraska Medical Center, Omaha, NE, United States of America, **2** Africa Centres for Disease Control and Prevention (Africa CDC), Addis Ababa, Ethiopia, **3** Amref Health Africa, Nairobi, Kenya

☯ These authors contributed equally to this work.
* nancy.tahmo@unmc.edu

## Abstract

Nigeria struggles with seasonal outbreaks of Lassa fever (LF), with 70 to 100% of its states affected annually. Since 2018, the seasonal dynamics have changed with a stark increase in infections, though the pattern in 2021 differed from the other years. Nigeria had three outbreaks of Lassa Fever in 2021. In that year, Nigeria also experienced substantial burdens from COVID-19 and Cholera. There is potential that these three outbreak events interacted with each other. This may have been from community disruption and so changes in how people access the health system, how the health system responds, or overlapping biological interactions, misclassification, social factors, misinformation, and pre-existing disparities and vulnerabilities. We assessed the syndemic potential of Lassa Fever, COVID-19, and Cholera through modeling their interactions across the 2021 calendar year employing a Poisson regression model. We included the number of states affected and the month of the year. We used these predictors to forecast the progression of the outbreak using a Seasonal Autoregressive Integrated Moving Average (SARIMA) model. The Poisson model prediction for the confirmed number of Lassa fever cases was significantly dependent on the number of confirmed COVID-19 cases, the number of states affected, and the month of the year (p-value < 0.001), and the SARIMA model was a good fit, accounting for 48% of the change in the number of cases of Lassa fever (p-value < 0.001) with parameters ARIMA (6, 1, 3) (5, 0, 3). Lassa Fever, COVID-19, and Cholera 2021 case curves have mirrored dynamics and likely interact. Further research into common, intervenable aspects of those interactions should be performed.

**Data Availability Statement:** All data were obtained through Nigeria Centre for Disease Control and Prevention disease situation reports

## Introduction

Concomitant epidemics in Nigeria are the norm. The first case of Lassa fever was reported in Nigeria in 1969 and the disease is endemic in West Africa, with regular case identification in Benin, Ghana, Guinea, Liberia, Mali, Sierra Leone, Togo, and Nigeria. An estimated 100,000 to

for 2021 (https://ncdc.gov.ng/diseases/sitreps/?cat=5&name=An%20update%20of%20Lassa%20fever%20outbreak%20in%20Nigeria).

**Funding:** The authors received no specific funding for this work.

**Competing interests:** The authors have declared that no competing interests exist.

300,000 cases and 5000 deaths occur annually [1–3]. On average, the case-fatality rate is 1% with a 15% case-fatality rate among those hospitalized [4]. Lassa fever is caused by a single-stranded RNA Arenavirus, Lassa virus (LASV), and transmission primarily occurs via direct contact with the droppings or urine of Mastomys rats, inhalation or ingestion of contaminated dust, or among humans, via contact with infected blood, tissue, or body fluids (secretions or excretions) such as urine or semen [1,5,6]. LASV still remains a priority pathogen for research and development, according to the World Health Organization (WHO), and resources have been committed to vaccine development [7]. The disease is a common nosocomial infection, particularly in settings where infection prevention and control practice is inhibited by staffing and supply shortage, including unavailable or improperly used personal protective equipment (PPE) [8,9].

Nigeria has experienced a steady increase in the number of confirmed Lassa fever cases; in 2018, 2019, and 2020, they were 633, 810, and 1189, respectively, across 29 out of 36 states. These numbers jumped to a cumulative 4654 suspected and 510 confirmed cases in 2021 recorded in 17 States, with a cumulative case fatality ratio of 20% [10]. Though the notable increase from 2020, disease statistics in 2021 were likely underreported, mainly due to the Severe Acute Respiratory Syndrome Coronavirus 2 (SARS-CoV-2 or COVID-19) pandemic, which strained the health system and pushed many to avoid seeking for medical attention.

Furthermore, shared epidemiologic and pathophysiologic features between Lassa fever, COVID-19, and Cholera add to challenges in appropriate case classification when laboratory confirmation is not attained [11]. In 2021, as well, Nigeria faced one of its worst cholera outbreaks in a decade, with 111,062 suspected cases in all 36 States, 3,604 of whom died [12,13]. There is potential that these three outbreak events interacted with each other. This may have been from community disruption and so changes in how people access the health system, how the health system responds, overlapping biological interactions, misclassification, social factors, misinformation, and pre-existing disparities and vulnerabilities.

We assessed the syndemic potential of Lassa fever, COVID-19, and cholera through modeling their interactions across the 2021 calendar year employing a Poisson regression model, including the number of States affected and the month of the year. We used these predictors to forecast the progression of the outbreak using a Seasonal Autoregressive Integrated Moving Average (SARIMA) model.

## Methods

The Poisson or Log-regression model was used to predict the frequency of confirmed cases of Lassa fever expressed as a function of the number of confirmed COVID-19 cases and the number of suspected cholera cases. We adjusted for the month of the year due to the seasonal variability of Lassa fever and the number of states affected as population size impacts disease persistence [14–16].

The SARIMA is a time-series machine learning model applied to forecast the incidence of infectious diseases [17–19]. This model controls for seasonality of the data and predicts based on previously reported cases, denoted by three paired parameters, ARIMA (p, d, q) × (P, D, Q) s, accounting for time lag and error. P = seasonal regression, D = seasonal differencing, Q = seasonal moving average coefficient (error). We use a partition of four seasons accounting for the average epidemiological weeks per month [20] relevant to the burden of disease.

Using data from the Nigeria Centre for Disease Control's Technical Guidelines for Integrated Disease Surveillance and Response (IDSR) and situation reports, we plotted epidemiological curves of confirmed Lassa fever, confirmed Covid-19, and suspected Cholera [21]. Case definitions, as used by Nigeria Centre for Disease Control, for the respective diseases are

excerpted in S1 Table, and the data in S1 Data. All statistical analyses were conducted using Statistical Package for the Social Sciences, SPSS IBM version 26 and a statistically significant value of 0.05 as reference [22].

## Results

The epidemiological curves for Lassa fever and COVID-19 appeared to be multi-modal, with two peaks around weeks seven to ten and one to five respectively (Fig 1). Fig 2 shows the epidemiological curve for Cholera, with a single peak around weeks 31–33 and an overlay of the suspected Cholera and confirmed COVID-19 case epidemiologic curves.

The Poisson model prediction for the confirmed number of Lassa fever cases was significantly dependent on the number of confirmed COVID-19 cases, number of States affected and the month of the year (p-value < 0.001) (Table 1).

After fitting the data with the number of confirmed COVID-19 cases, number of States affected and the month of the year as predictors, Table 2 and Fig 3 show that the model for predicting Lassa fever cases is a good fit. The model accounts for 48.1% of change in the number of cases of Lassa fever (p-value 0.001) over 2021, with the SARIMA model parameters ARIMA (6, 1, 3) (5, 0, 3) (Table 2).

## Discussion and conclusion

Generally, cases peaked between January and March when respiratory diseases are most common (rainy season) and thus, a temporal coincidence with the COVID-19 peak in Nigeria [23–25]. Outcome ascertainment is very prone to bias. The incubation period of Lassa Fever ranges between 2 to 21 days, with a wide spectrum of clinical manifestations, many of which are non-specific (fever, headache, general weakness).They progress after a few days to mild symptoms (sore throat, muscle pain, nausea, vomiting, chest pain, arthralgia, diarrhea) and, in some cases, may progress to a range of complications (respiratory distress, multiorgan failure, hepatitis, bleeding from mucosa surfaces–mouth, vagina, or gastrointestinal tract, encephalopathy, or spontaneous abortions). About 80% of cases are asymptomatic and subclinical and those who recover can potentially get reinfected. [14,26–28].

Several computational and mathematical models have been developed to describe or predict the transmission dynamics and spatial trends of Lassa fever, employing different predictors such as the number of susceptible, infected, exposed, or recovered persons (theoretical model-

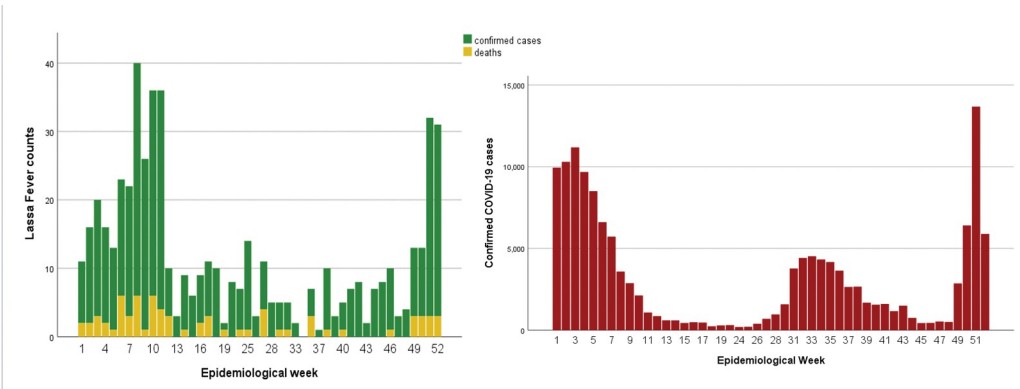

**Fig 1. Epidemiologic curves for Lassa fever and COVID-19 confirmed cases, Nigeria 2021.**

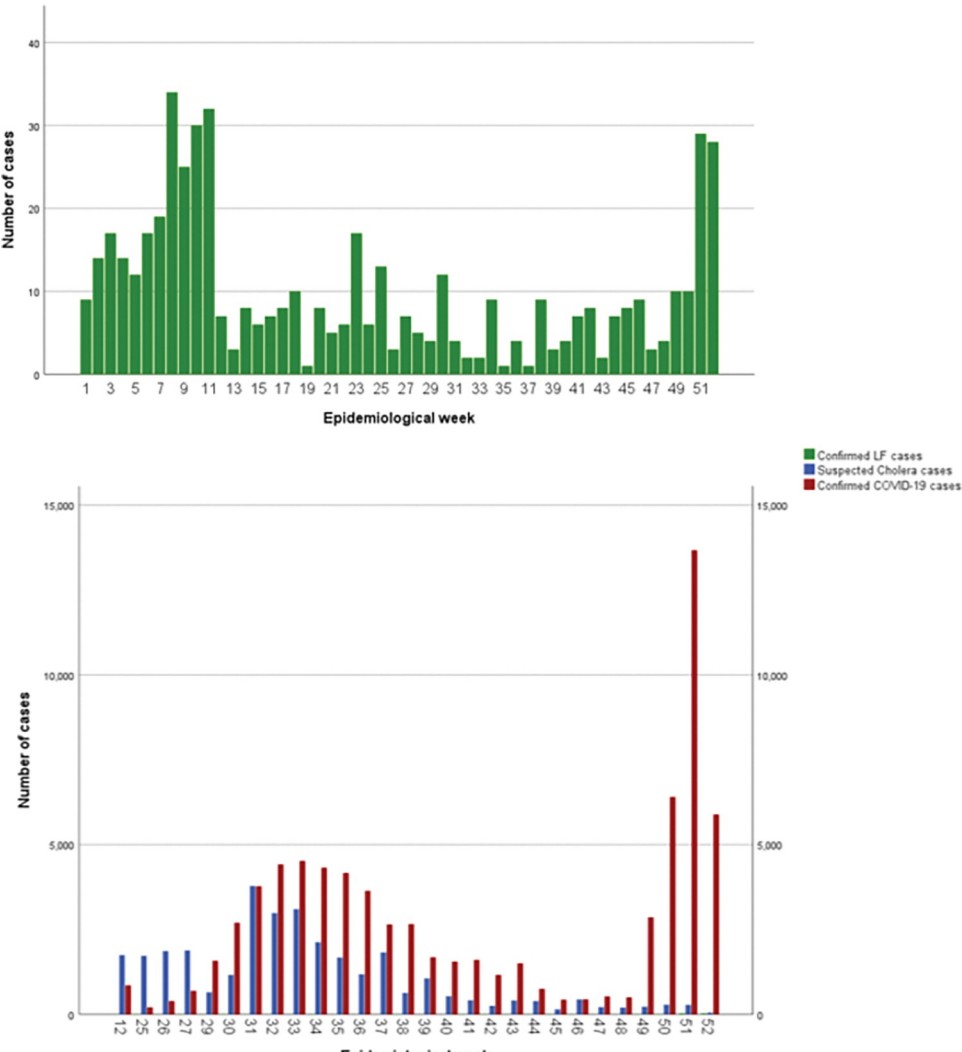

**Fig 2. Overlay and panel of confirmed Lassa fever and COVID-19 and suspected Cholera cases epidemiological curves, Nigeria 2021.**

SIERS), climatic factors, socioeconomic status, seroprevalence, or hospitalization, fitted to real data from epidemics [29–32].

Literature exists documenting major driving forces of Lassa fever outbreaks and transmission dynamics, such as the seasonal migration of rodent populations and their consumption,

**Table 1. Poisson regression model estimates for Lassa fever cases for different covariates.**

| Variable | B | 95% CI | Exp(B) | 95% CI | p-value |
|---|---|---|---|---|---|
| Month of the year | -0.135 | -0.234, -0.036 | 0.874 | 0.791, 0.965 | <0.001 |
| Suspected Cholera cases | 0.000 | 0.000, 0.000 | 0.000 | 1.000, 1.000 | 0.237 |
| Confirmed COVID-19 cases | 0.000 | 0.000, 0.000 | 0.000 | 1.000, 1.000 | <0.001 |
| Number of States | 0.412 | 0.262, 0.562 | 1.510 | 1.300, 1.755 | <0.001 |

**Notes.** B: Unstandardized Beta estimate; Exp(B): Exponent of Beta estimates.

**Table 2. SARIMA model estimates.**

| Model | Number of predictors | Model fit R-squared | p-value |
|---|---|---|---|
| Confirmed Lassa fever cases | 3 | 0.481 | 0.001 |

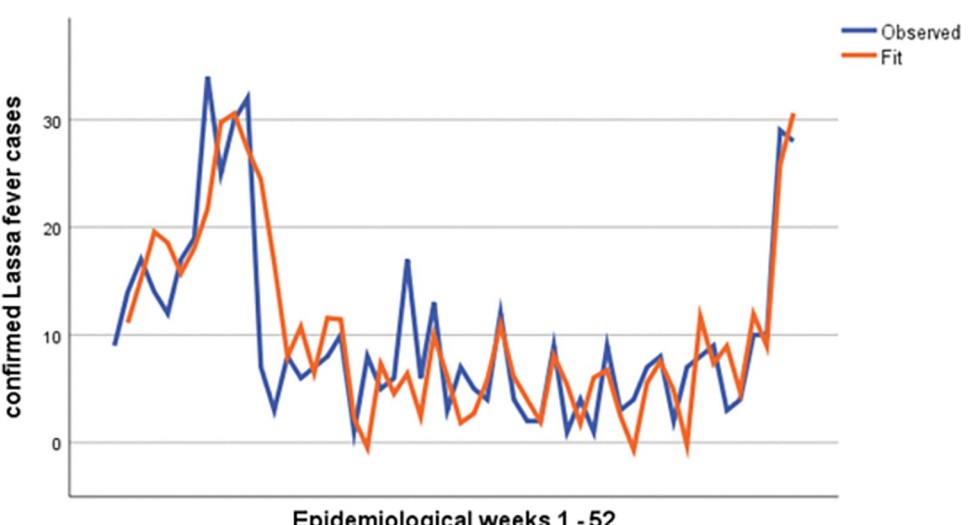

**Fig 3. SARIMA forecast plot.**

especially around forested areas [28,33], geography, and climatic conditions [34]. This study sought to explore concomitant epidemics as transmission drivers, especially with the potential biological or co-infection effects (host factors) on Nigerians who in 2021 faced three epidemics. The diseases caused by Lassa virus, SARS-CoV2, and *Vibrio cholerae* share some clinical manifestations, and many cases of Lassa fever and COVID-19 are asymptomatic [35]. In addition, before the COVID-19 pandemic, the Nigerian health ministry struggled with an under-resourced health infrastructure serving about 36 States [11]. This study identifies interactions between Nigeria's 2021 Lassa fever, COVID-19, and cholera epidemics. Resource shifting public health programming, community disruption leading to increased exposures, health system strain, and even co-infection and biologic factors all may play a part in these findings. Further research is merited. In particular, it is important to create more awareness and improve resilient surveillance and risk assessment mechanisms that provide actionable information even when in the midst of multiple communicable disease epidemics. A major challenge with this analysis was the absence of specific cholera cases prior to epidemiologic week 12, together with better ratio of confirmed to suspect cases, which would improve the value of the output from the SARIMA model.

## Supporting information

**S1 Table. Includes case definitions used for case classification.**
(DOCX)

**S1 Data. The data used for this research.**
(DOCX)

## Acknowledgments

We thank Sarah Sanasac who reviewed and provided valuable comments on the first draft of this work.

## Author Contributions

**Conceptualization:** Nancy B. Tahmo, David M. Brett-Major.

**Data curation:** Nancy B. Tahmo.

**Formal analysis:** Nancy B. Tahmo.

**Funding acquisition:** David M. Brett-Major.

**Investigation:** Nancy B. Tahmo, Frankline Sevidzem Wirsiy.

**Methodology:** Nancy B. Tahmo, Frankline Sevidzem Wirsiy.

**Project administration:** David M. Brett-Major.

**Software:** Nancy B. Tahmo.

**Supervision:** David M. Brett-Major.

**Validation:** Frankline Sevidzem Wirsiy.

**Visualization:** Nancy B. Tahmo.

**Writing – original draft:** Nancy B. Tahmo.

**Writing – review & editing:** Frankline Sevidzem Wirsiy, David M. Brett-Major.

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
