## [Decision Letter · Decision Letter 0]

15 Mar 2023

PGPH-D-23-00263

Modeling the Lassa fever outbreak synchronously occurring with cholera and COVID-19 outbreaks in Nigeria 2021: a threat to Global Health Security

Dear Dr. Tahmo,

Thank you for submitting your manuscript to PLOS Global Public Health. After careful consideration, we feel that it has merit but does not fully meet PLOS Global Public Health’s publication criteria as it currently stands. Therefore, we invite you to submit a revised version of the manuscript that addresses the points raised during the review process.

EDITOR: Please insert comments here and delete this placeholder text when finished. Be sure to:

Indicate which changes you require for acceptance versus which changes you recommendAddress any conflicts between the reviews so that it's clear which advice the authors should followProvide specific feedback from your evaluation of the manuscript

Please ensure that your decision is justified on PLOS Global Public Health’s publication criteria and not, for example, on novelty or perceived impact.

We look forward to receiving your revised manuscript.

Kind regards,

Saskia Popescu, PhD

Academic Editor

Journal Requirements:

Additional Editor Comments (if provided):

Please provide details on the disease surveillance tools employed to detect cases of the individual diseases with context about the reliability of case data.

Reviewers' comments:

Reviewer's Responses to Questions

**Comments to the Author**

1. Does this manuscript meet PLOS Global Public Health’s publication criteria? Is the manuscript technically sound, and do the data support the conclusions? The manuscript must describe methodologically and ethically rigorous research with conclusions that are appropriately drawn based on the data presented.

Reviewer #1: Yes

2. Has the statistical analysis been performed appropriately and rigorously?

Reviewer #1: Yes

3. Have the authors made all data underlying the findings in their manuscript fully available (please refer to the Data Availability Statement at the start of the manuscript PDF file)?

Reviewer #1: Yes

4. Is the manuscript presented in an intelligible fashion and written in standard English?

Reviewer #1: Yes

5. Review Comments to the Author

Reviewer #1: I recommend that this paper be accepted after minor revision. Please provide details on the disease surveillance tools employed to detect cases of the individual diseases with context about the reliability of case data.

6. PLOS authors have the option to publish the peer review history of their article (what does this mean?). If published, this will include your full peer review and any attached files.

**Do you want your identity to be public for this peer review?** For information about this choice, including consent withdrawal, please see our Privacy Policy.

Reviewer #1: **Yes: **Jessica Malaty Rivera, MS

---

## [Editor Report · Decision Letter 1]

25 Apr 2023

Modeling the Lassa fever outbreak synchronously occurring with cholera and COVID-19 outbreaks in Nigeria 2021: a threat to Global Health Security

PGPH-D-23-00263R1

Dear Tahmo,

We are pleased to inform you that your manuscript 'Modeling the Lassa fever outbreak synchronously occurring with cholera and COVID-19 outbreaks in Nigeria 2021: a threat to Global Health Security' has been provisionally accepted for publication in PLOS Global Public Health.

Best regards,

Saskia Popescu, PhD

Academic Editor